# INSTRUCTG2I: Synthesizing Images from Multimodal Attributed Graphs

**Bowen Jin, Ziqi Pang, Bingjun Guo, Yu-Xiong Wang, Jiaxuan You, Jiawei Han**
Department of Computer Science
University of Illinois at Urbana-Champaign
bowenj4@illinois.edu
https://instructg2i.github.io/

## Abstract

In this paper, we approach an overlooked yet critical task *Graph2Image*: generating images from multimodal attributed graphs (MMAGs). This task poses significant challenges due to the explosion in graph size, dependencies among graph entities, and the need for controllability in graph conditions. To address these challenges, we propose a graph context-conditioned diffusion model called INSTRUCTG2I. INSTRUCTG2I first exploits the graph structure and multimodal information to conduct informative neighbor sampling by combining personalized page rank and re-ranking based on vision-language features. Then, a Graph-QFormer encoder adaptively encodes the graph nodes into an auxiliary set of *graph prompts* to guide the denoising process of diffusion. Finally, we propose graph classifier-free guidance, enabling controllable generation by varying the strength of graph guidance and multiple connected edges to a node. Extensive experiments conducted on three datasets from different domains demonstrate the effectiveness and controllability of our approach. The code is available at https://github.com/PeterGriffinJin/InstructG2I.

## 1 Introduction

This paper investigates an overlooked yet critical source of information for image generation: the pervasive *graph-structured relationships* of real-world entities. In contrast to the commonly adopted language conditioning in models represented by Stable Diffusion [32], graph connections have *combinatorial complexity* and cannot be trivially captured as a sequence. Such graph-structured relationships among the entities are expressed through "*Multimodal Attributed Graphs*" (MMAGs), where nodes are enriched with image and text information. As a concrete example (Figure 1(a)), the graph of artworks is constructed by nodes containing images (pictures) and texts (titles), as well as edges corresponding to shared genre and authorship. Such a graph uniquely depicts a piece of artwork by its thousands of peers in the graph, beyond the mere description of language.

To this end, we formulate and propose the *Graph2Image* challenge, requiring the generative models to synthesize image conditioning on both text descriptions and graph connections of a node. This task featuring the image generation on MMAGs is well-grounded in real-world applications. For instance, generating an image for a virtual artwork node in the art MMAG is akin to creating virtual artwork according to the nuanced styles of artists and genres [5] (as in Figure 1(a)). Similarly, generating an image for a product node connected to other products through co-purchase links in an e-commerce MMAG equates to recommending future products for users [24]. Without surprise, our exploiting the graph-structured information indeed improves the consistency of generated images compared to models only using texts or images as conditioning (Figure 1(b)).

38th Conference on Neural Information Processing Systems (NeurIPS 2024).

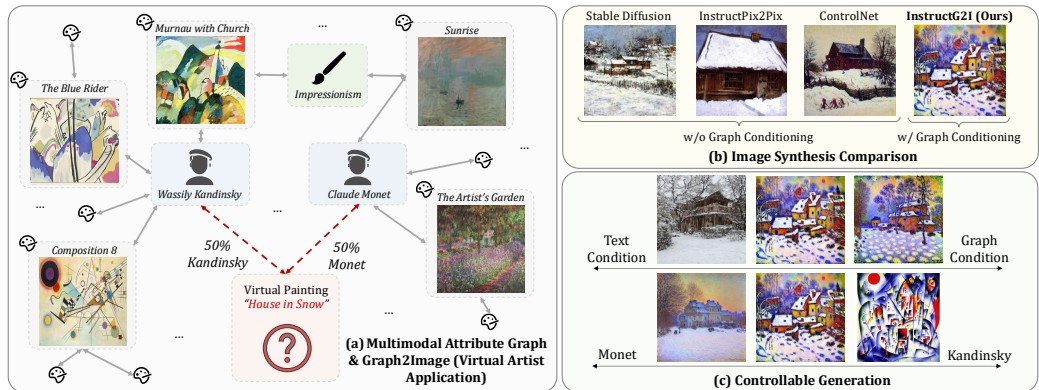

Figure 1: We propose a new task *Graph2Image* featuring image synthesis by conditioning on graph information and introduce a novel graph-conditioned diffusion model called INSTRUCTG2I to tackle this problem. (a) *Graph2Image* is supported by prevalent multimodal attributed graphs and is grounded in real-world applications, *e.g.*, virtual artistry. (b) INSTRUCTG2I outperforms baseline image generation techniques, demonstrating the usefulness of graph information. (c) To accommodate realistic user queries, INSTRUCTG2I exhibits smooth controllability in utilizing text/graph information and managing the strength of multiple graph edges.

Despite the usefulness of graph information, existing methods conditioning on either text [32] or images [2, 41] are incapable of direct integration with MMAGs. Therefore, we propose a graph context-aware diffusion model INSTRUCTG2I inherited from Stable Diffusion that mitigates gaps. A most prominent challenge directly originates from the combinatorial complexity of graphs, which we term as *Graph Size Explosion*: inputting the entire local subgraph structure to a model, including all the images and texts, is impractical due to the exponential increase in size, especially with additional hops. Therefore, INSTRUCTG2I learns to *compress* the massive amounts of contexts from the graph into a set of *graph conditioning* tokens with fixed capacity, which functions alongside the common text conditioning tokens in Stable Diffusion. Such a compression process is enhanced with a *semantic personalized pagerank-based graph sampling* approach to actively select the most informative neighboring nodes based on both structural and semantic perspectives.

Besides the *number* of contexts, the graph structures in MMAGs additionally specify the proximity of entities, which is not captured in conventional text or image conditioning. This challenge of "*Graph Entity Dependency*" reflects the implicit preference of image generation: synthesizing a shirt image linked to "light-colored" clothing is likely to have a "pastel tone" (image-image dependency), and generating a picture titled "a running horse" should reference interconnected animal images rather than scenic ones (text-image dependency). To enable the nuanced proximity understanding on graphs, we further improve our graph conditioning tokens via a Graph-QFormer architecture learning to encode the graph information guided by texts.

Finally, we propose that our graph conditioning is a natural interface for *controllable* generation, reflecting the strength of edges in MMAGs. Take the virtual art generation (Figure 1(c)) for example: INSTRUCTG2I can flexibly offer different strengths of graph guidance and can smoothly transition between the style of Monet and Kandinsky, defined by its strength of connection with either of the two artists. Such an advantage is grounded for real-world application and is a *plug-and-play* test-time algorithm inspired by classifier-free guidance [18]. In sum, our contributions include:

- *Formulation and Benchmark*. We are the first to identify the usefulness of multimodal attributed graphs (MMAGs) in image synthesis and formulate the *Graph2Image* problem. Our formulation is supported by three benchmarks grounded in the real-world applications of art and e-commerce.

- *Algorithm*. Methodologically, we propose INSTRUCTG2I, a context-aware diffusion model that effectively encodes graph conditional information as graph prompts for controllable image generation (as shown in Figure 1(b,c)).

- *Experiments and Evaluation*. Empirically, we conduct experiments on graphs from three different domains, demonstrating that INSTRUCTG2I consistently outperforms competitive baselines (as shown in Figure 1(b)).

## 2  Problem Formulation

### 2.1  Multimodal Attributed Graphs

**Definition 1** *(Multimodal Attributed Graphs (MMAGs))* A multimodal attributed graph can be defined as $\mathcal{G} = (\mathcal{V}, \mathcal{E}, \mathcal{P}, \mathcal{D})$, where $\mathcal{V}, \mathcal{E}, \mathcal{P}$ and $\mathcal{D}$ represent the sets of nodes, edges, images, and documents, respectively. Each node $v_i \in \mathcal{V}$ is associated with some textual information $d_{v_i} \in \mathcal{D}$ and some image information $p_{v_i} \in \mathcal{P}$.

For example, in an e-commerce product graph, nodes ($v \in \mathcal{V}$) represent products, edges ($e \in \mathcal{E}$) denote co-viewed semantic relationships, images ($p \in \mathcal{P}$) are product images, and documents ($d \in \mathcal{D}$) are product titles. Similarly, in an art graph (shown in Figure 1), nodes represent artworks, edges signify shared artists or genres, images are artwork pictures, and documents are artwork titles.

In this work, we focus on graphs where edges provide *semantic correlations* between images (nodes). For instance, in an e-commerce product graph, connected products (those frequently co-viewed by many users) are highly related. Similarly, in an art graph, linked artworks (those created by the same author or within the same genre) are likely to have similar styles.

### 2.2  Problem Definition

In this work, we explore the problem of node image generation on MMAGs. Given a node $v_i$ in an MMAG $\mathcal{G}$, our objective is to generate $p_{v_i}$ based on $d_{v_i}$ and $\mathcal{G}$. This problem has multiple real-world applications. For example, in e-commerce, it translates to generating the image ($p_{v_i}$) for a product ($v_i$) based on a user query ($d_{v_i}$) and user purchase history ($\mathcal{G}$), which is a generative retrieval task. In the art domain, it involves generating the picture ($p_{v_i}$) for an artwork ($v_i$) based on its title ($d_i$) and its associated artist style or genre ($\mathcal{G}$), which is a virtual artwork creation task.

**Definition 2** *(Node Image Generation on MMAGs)* In a multimodal attributed graph $\mathcal{G} = (\mathcal{V}, \mathcal{E}, \mathcal{P}, \mathcal{D})$, given a node $v_i \in \mathcal{V}$ within the graph $\mathcal{G}$ with a textual description $d_{v_i}$, the goal is to synthesize $p_{v_i}$, the corresponding image at $v_i$, with a learned model $\hat{p}_{v_i} = f(v_i, d_{v_i}, \mathcal{G})$.

Our evaluation emphasizes instance-level similarity, assessing how closely $\hat{p}_{v_i}$ matches $p_{v_i}$. We conduct evaluations on artwork graphs, e-commerce graphs, and literature graphs. More details can be found in Section 4.1.

## 3  Methodology

In this section, we present our INSTRUCTG2I framework, overviewed in Figure 2. We begin by introducing graph conditions into stable diffusion models in Section 3.1. Next, we discuss semantic personalized PageRank-based sampling to select informative graph conditions in Section 3.2. Furthermore, we propose Graph-QFormer to extract dependency-aware representations for graph conditions in Section 3.3. Finally, we introduce controllable generation to balance the condition scale between text and graph guidance, as well as manage multiple graph guidances in Section 3.4.

### 3.1  Graph Context-aware Stable Diffusion

**Stable Diffusion (SD).** INSTRUCTG2I is built upon Stable Diffusion (SD). SD conducts diffusion in the latent space, where an input image $x$ is first encoded from pixel space into a latent representation $\mathbf{z} = \text{Enc}(x)$. A decoder then transfers the latent representation $\mathbf{z}'$ back to the pixel space, yielding $x' = \text{Dec}(\mathbf{z}')$. The diffusion model generates the latent representation $\mathbf{z}'$ conditioned on a text prompt $c_T$. The training objective of SD is defined as follows:

$$\mathcal{L} = \mathbb{E}_{\mathbf{z} \sim \text{Enc}(x), c_T, \epsilon \sim \mathcal{N}(0,1), t} \left[ \| \epsilon - \epsilon_\theta(\mathbf{z}_t, t, h(c_T)) \|^2 \right]. \tag{1}$$

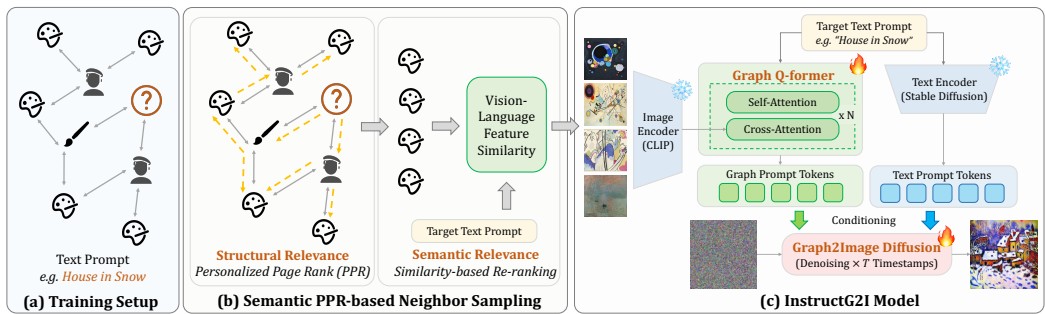

Figure 2: The overall framework of INSTRUCTG2I. (a) Given a target node with a text prompt (*e.g.*, *House in Snow*) in a Multimodal Attributed Graph (MMAG) for which we want to generate an image, (b) we first perform semantic PPR-based neighbor sampling, which involves structure-aware personalized PageRank and semantic-aware similarity-based reranking to sample informative neighboring nodes in the graph. (c) These neighboring nodes are then inputted into a Graph-QFormer, encoded by multiple self-attention and cross-attention layers, represented as *graph tokens* and used to guide the denoising process of the diffusion model, together with text prompt tokens.

At each timestep $t$, the denoising network $\epsilon_\theta(\cdot)$ predicts the noise by conditioning on the current latent representation $\mathbf{z}_t$, timestep $t$ and text prompt vectors $h(c_T)$. To compute $h(c_T) \in \mathbf{R}^{d \times l_{c_T}}$, where $l_{c_T}$ is the length of $c_T$ and $d$ is the hidden dimension, the text prompt $c_T$ is processed by the CLIP text encoder [31]: $h(c_T) = \text{CLIP}(c_T)$.

**Introducing Graph Conditions into SD.** In the context of MMAGs, synthesizing the image for a node $v_i$ involves not only the text $d_{v_i}$, but also the semantic information from the node's proximity on the graph. Therefore, we introduce an auxiliary set of *graph conditioning tokens* $h_G(c_G)$ to the SD models (as shown in Figure 2(c)), working in parallel with the existing text conditions $h_T(c_T)$.

$$h(c_T, c_G) = [h_T(c_T), h_G(c_G)] \in \mathbf{R}^{d \times (l_{c_T} + l_{c_G})}, \tag{2}$$

where $l_{c_G}$ is the length of the graph condition. The training objective then becomes:

$$\mathcal{L} = \mathbb{E}_{\mathbf{z} \sim \text{Enc}(x), c_T, c_G, \epsilon \sim \mathcal{N}(0,1), t} \left[ \| \epsilon - \epsilon_\theta(\mathbf{z}_t, t, h(c_T, c_G)) \|^2 \right]. \tag{3}$$

For $h_T(c_T)$, we can directly use the CLIP text encoder as in the original SD. However, determining $c_G$ and $h_G(\cdot)$ is more complex. We will discuss the details of $c_G$ and $h_G(\cdot)$ in the following sections.

## 3.2 Semantic PPR-based Neighbor Sampling

A straightforward approach to developing $c_G(v_i)$ involves using the entire local subgraph of $v_i$. However, this is impractical due to the exponential growth in size with each additional hop, leading to excessively long context sequences. To address this, we leverage both graph structure and node semantics to select informative $c_G$.

**Structure Proximity: Personalized PageRank (PPR)**. Inspired by [10], we first adopt PPR [15] to identify related nodes from a graph structure perspective. PPR processes the graph structure to derive a ranking score $P_{i,j}$ for each node $v_j$ relative to node $v_i$, where a higher $P_{i,j}$ indicates a greater degree of "similarity" between $v_i$ and $v_j$. Let $\boldsymbol{P} \in \mathbf{R}^{n \times n}$ be the PPR matrix of the graph, where each row $P_{i,:}$ represents a PPR vector toward a target node $v_i$. The matrix $\boldsymbol{P}$ is determined by:

$$\boldsymbol{P} = \beta \hat{\boldsymbol{A}} \boldsymbol{P} + (1 - \beta) \boldsymbol{I}. \tag{4}$$

where $\beta$ is the reset probability for PPR and $\hat{\boldsymbol{A}}$ is the normalized adjacency matrix. Once $\boldsymbol{P}$ is computed, we define the PPR-based graph condition $c_{G_{\text{ppr}}}$ of node $v_i$ as the top-$K_{\text{ppr}}$ PPR neighbors of node $v_i$:

$$c_{G_{\text{ppr}}}(v_i) = \underset{c_{G_{\text{ppr}}}(v_i) \subset \mathcal{V}, |c_{G_{\text{ppr}}}(v_i)| = K_{\text{ppr}}}{\text{argmax}} \sum_{v_j \in c_{G_{\text{ppr}}}(v_i)} P_{i,j}. \tag{5}$$

**Semantic Proximity: Similarity-based Reranking**. However, solely relying on PPR may result in a graph condition set containing images (*e.g.,* scenery pictures) that are not semantically related to our

target node (*e.g.,* a picture titled "running horse"). To address this, we propose using a semantic-based similarity calculation function $\text{Sim}(d, p)$ (*e.g.*, CLIP) to rerank $v_j \in c_{G_{\text{ppr}}}(v_i)$ based on the relatedness of $p_{v_j}$ to $d_{v_i}$. The final graph condition $c_G(v_i)$ is calculated by:

$$c_G(v_i) = \underset{c_G(v_i) \subset c_{G_{\text{ppr}}}(v_i), |c_G(v_i)|=K}{\text{argmax}} \sum_{v_j \in c_G(v_i)} \text{Sim}(d_{v_i}, p_{v_j}). \tag{6}$$

### 3.3 Graph Encoding with Text Conditions

After we derive $c_G(v_i)$ from the previous step, the problem comes to how can we design $h_G(\cdot)$ to extract meaningful representations from $c_G(v_i)$. Here we focus more on how to utilize the image features from $c_G(v_i)$ (*i.e.*, $\{p_{v_j}|v_j \in c_G(v_i)\}$) since we find they are more informative for $v_i$ image generation compared with text features from $c_G(v_i)$ (*i.e.*, $\{d_{v_j}|v_j \in c_G(v_i)\}$) (shown in Section 4.3).

**Simple Baseline: Encoding with Pretrained Image Encoders [31].** A straightforward way to obtain representations for $v_j \in c_G(v_i)$ is to directly apply some pretrained image encoders $g_{\text{img}}(\cdot)$ (*e.g.*, CLIP [31]):

$$\boldsymbol{h}_{v_j} = g_{\text{img}}(p_{v_j}) \in \mathbf{R}^d, \ \ h_G(c_G(v_i)) = \oplus[\boldsymbol{h}_{v_j}]_{v_j \in c_G(v_i)} \in \mathbf{R}^{d \times l_{c_G}}, \tag{7}$$

where $\oplus$ denotes the concatenation operation. However, this simple design has two significant limitations: 1) The encoding for each $p_{v_j}$ ($v_j \in c_G(v_i)$) is isolated from others in $c_G(v_i)$ and failed to capture the image-image graph dependency. For example, the style extraction from one picture ($p_{v_j}$) can benefit from the other pictures created by the same artist (in $c_G(v_i)$). 2) The encoding for each $p_{v_j}$ is independent to $d_{v_i}$, which fails to capture the text-image graph dependency. For example, when we are creating a picture titled "running horse" ($d_{v_i}$), it is desired to offer more weight on horse pictures in $c_G(v_i)$ rather than scenery pictures.

**Graph-QFormer.** To address these limitations, we propose Graph-QFormer as $h_G(\cdot)$ to learn representations for $c_G$ while considering the graph dependency information. As shown in Figure 2, Graph-QFormer consists of two Transformer [35] modules motivated by [26]: (1) a self-attention module that facilitates deep mutual information exchange between previous layer hidden states, capturing image-image dependencies and (2) a cross-attention module that weights samples in $c_G$ using text guidance, capturing text-image dependencies.

Let $\boldsymbol{H}_{c_G(v_i)}^{(t)} \in \mathbf{R}^{d \times l_{c_G}}$ denote the hidden states outputted by the $t$-th Graph-QFormer layer. We use the token embeddings of $d_{v_i}$ as the input query embeddings to provide text guidance:

$$\boldsymbol{H}_{c_G(v_i)}^{(0)} = [\boldsymbol{x}_1, ..., \boldsymbol{x}_{|d_{v_i}|}]. \tag{8}$$

where $\boldsymbol{x}_k$ is the $k$-th token embedding in $d_{v_i}$ and $l_{c_G} = |d_{v_i}|$. The multi-head self-attention layer ($\text{MHA}_{\text{SAT}}$) is calculated by

$$\boldsymbol{H'}_{c_G(v_i)}^{(t)} = \text{MHA}_{\text{SAT}}[q = \boldsymbol{H}_{c_G(v_i)}^{(t-1)}, k = \boldsymbol{H}_{c_G(v_i)}^{(t-1)}, v = \boldsymbol{H}_{c_G(v_i)}^{(t-1)}], \tag{9}$$

where $q, k, v$ denotes query, key, and value channels in the Transformer. The output $\boldsymbol{H'}_{c_G(v_i)}^{(t)}$ is then inputted to the multi-head cross-attention layer ($\text{MHA}_{\text{CAT}}$), calculated by

$$\boldsymbol{H}_{c_G(v_i)}^{(t)} = \text{MHA}_{\text{CAT}}[q = \boldsymbol{H'}_{c_G(v_i)}^{(t)}, k = \boldsymbol{Z}_{c_G(v_i)}, v = \boldsymbol{Z}_{c_G(v_i)}], \tag{10}$$

where $\boldsymbol{Z}_{c_G(v_i)} = \oplus[g_{\text{img}}(p_{v_j})]_{v_j \in c_G(v_i)} \in \mathbf{R}^{d \times n}$ represents the image embeddings extracted from a fixed pretrained image encoder and $n$ is the number of embeddings. Finally we adopt $h_G(c_G(v_i)) = \boldsymbol{H}_{c_G(v_i)}^{(L)}$, where $L$ is the number of layers in Graph-QFormer.

**Connection between INSTRUCTG2I and GNNs.** As illustrated in Figure 2, INSTRUCTG2I employs a Transformer-based architecture as the graph encoder. However, it can also be interpreted as a Graph Neural Network (GNN) model. GNN models [38] primarily use a propagation-aggregation paradigm to obtain node representations ($\mathcal{N}(i)$ denotes the neighbor set of $i$):

$$\boldsymbol{a}_{ij}^{(l-1)} = \text{PROP}^{(l)}\left(\boldsymbol{h}_i^{(l-1)}, \boldsymbol{h}_j^{(l-1)}\right), (\forall j \in \mathcal{N}(i)); \ \ \boldsymbol{h}_i^{(l)} = \text{AGG}^{(l)}\left(\boldsymbol{h}_i^{(l-1)}, \{\boldsymbol{a}_{ij}^{(l-1)}|j \in \mathcal{N}(i)\}\right).$$

Similarly, in INSTRUCTG2I, Eq.(4)(5)(6) can be regarded as the propagation function $\text{PROP}^{(l)}$, while the aggregation step $\text{AGG}^{(l)}$ corresponds to the combination of Eq.(9) and Eq.(10).

## 3.4 Controllable Generation

The concept of classifier-free guidance, introduced by [18], enhances the performance of conditional image synthesis by modifying the noise prediction, $e_\theta(\cdot)$, with the output from an unconditional model. This is formulated as: $\hat{\epsilon}_\theta(\mathbf{z}_t, c) = \epsilon_\theta(\mathbf{z}_t, \varnothing) + s \cdot (\epsilon_\theta(\mathbf{z}_t, c) - \epsilon_\theta(\mathbf{z}_t, \varnothing))$, where $s(> 1)$ is the guidance scale. The intuition is that $\epsilon_\theta$ learns the gradient of the log image distribution and increasing the contribution of $\epsilon_\theta(c) - \epsilon_\theta(\varnothing)$ will enlarge the convergence to the distribution conditioned on $c$.

In our task, the score network $\hat{\epsilon}_\theta(\mathbf{z}_t, c_G, c_T)$ is conditioned on both text $c_T = d_i$ and the graph condition $c_G$. We compose the score estimates from these two conditions and introduce two guidance scales, $s_T$ and $s_G$, to control the contribution strength of $c_T$ and $c_G$ to the generated samples respectively. Our modified score estimation function is:

$$\hat{\epsilon}_\theta(\mathbf{z}_t, c_G, c_T) = \epsilon_\theta(\mathbf{z}_t, \varnothing, \varnothing) + s_T \cdot (\epsilon_\theta(\mathbf{z}_t, \varnothing, c_T) - \epsilon_\theta(\mathbf{z}_t, \varnothing, \varnothing))$$
$$+ s_G \cdot (\epsilon_\theta(\mathbf{z}_t, c_G, c_T) - \epsilon_\theta(\mathbf{z}_t, \varnothing, c_T)). \qquad (11)$$

For cases requiring fine-grained control over multiple graph conditions (*i.e.*, different edges), we extend the formula as follows:

$$\hat{\epsilon}_\theta(\mathbf{z}_t, c_G, c_T) = \epsilon_\theta(\mathbf{z}_t, \varnothing, \varnothing) + s_T \cdot (\epsilon_\theta(\mathbf{z}_t, \varnothing, c_T) - \epsilon_\theta(\mathbf{z}_t, \varnothing, \varnothing))$$
$$+ \sum s_G^{(k)} \cdot (\epsilon_\theta(\mathbf{z}_t, c_G^{(k)}, c_T) - \epsilon_\theta(\mathbf{z}_t, \varnothing, c_T)), \qquad (12)$$

where $c_G^{(k)}$ is the $k$-th graph condition. For example, to create an artwork that combines the styles of Monet and Van Gogh, the neighboring artworks by Monet and Van Gogh on the graph would be $c_G^{(1)}$ and $c_G^{(2)}$, respectively. Further details on the derivation of our classifier-free guidance formulations can be found in Appendix A.3.

# 4 Experiments

## 4.1 Experimental Setups

**Datasets.** We conduct experiments on three MMAGs from distinct domains: ART500K [27], Amazon [16], and Goodreads [37]. ART500K is an artwork graph with nodes representing artworks and edges indicating same-author or same-genre relationships. Each artwork node includes a title (text) and a picture (image). Amazon is a product graph where nodes represent products and edges denote co-view relationships. Each product is associated with a title (text) and a picture (image). Goodreads is a literature graph where nodes represent books and edges convey similar-book semantics. Each book node contains a title and a front cover image. Dataset statistics can be found in Appendix A.4.

**Baselines.** We compare INSTRUCTG2I with two groups of baselines: 1) Text-to-image methods: This includes Stable Diffusion 1.5 (SD-1.5) [32] and SD 1.5 fine-tuned on our datasets (SD-1.5 FT). 2) Image-to-image methods: This includes InstructPix2Pix [2] and ControlNet [41], both initialized with SD 1.5 and fine-tuned on our datasets. We use the most relevant neighbor, as selected in Section 3.2 as the input image for these baselines, allowing them to partially utilize graph information.

**Metrics.** As indicated in Section 2.2, our evaluation mainly concerns the consistency of synthesized images with the ground truth image on the node. Therefore, our evaluation adopts the CLIP [31] and DINOv2 [29] score for instance-level similarity, in addition to the conventional FID [17] metric for image generation. For the CLIP and DINOv2 scores, we utilize CLIP and DINOv2 to obtain representations for both the generated and ground truth images and then calculate their cosine similarity. For FID, we calculate the distance between the distribution of the ground truth images and the distribution of the generated images.

## 4.2 Main results

**Quantitative Evaluation.** The quantitative results are presented in Table 1 and Figure 3. From Table 1, we observe the following: 1) INSTRUCTG2I consistently outperforms all the baseline methods, highlighting the importance of graph information in image synthesis on MMAGs. 2) Although InstructPix2Pix and ControlNet partially consider graph context, they fail to capture the semantic signals from the graph comprehensively. In Figure 3, we plot the aver-

Table 1: Quantitative evaluation of different methods on ART500K, Amazon, and Goodreads datasets. The CLIP score denotes the image-image score. **INSTRUCTG2I significantly outperforms the best baseline** with p-value < 0.05 and consistently outperforms all the other common baselines in image synthesis, supporting the benefits of graph conditioning.

| Model | ART500K | | Amazon | | Goodreads | |
|---|---|---|---|---|---|---|
| | CLIP score | DINOv2 score | CLIP score | DINOv2 score | CLIP score | DINOv2 score |
| SD-1.5 | 58.83 | 25.86 | 60.67 | 32.61 | 42.16 | 14.84 |
| SD-1.5 FT | 66.55 | 34.65 | 65.30 | 41.52 | 45.81 | 18.97 |
| InstructPix2Pix | 65.66 | 33.44 | 63.86 | 41.31 | 47.30 | 20.94 |
| ControlNet | 64.93 | 32.88 | 59.88 | 34.05 | 42.20 | 19.77 |
| INSTRUCTG2I | **73.73** | **46.45** | **68.34** | **51.70** | **50.37** | **25.54** |

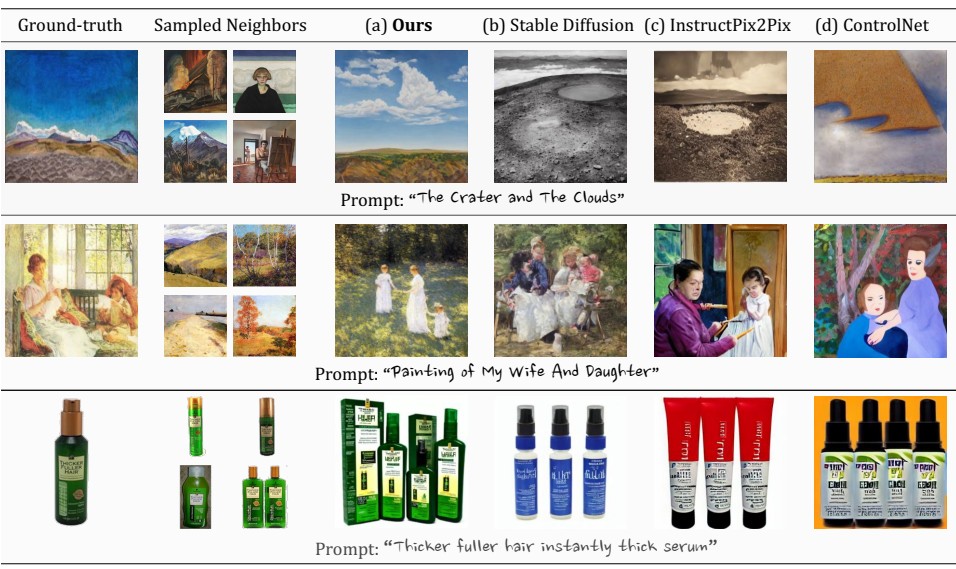

Figure 4: Qualitative evaluation. **Our method exhibits better consistency with the ground truth** by better utilizing the graph information from neighboring nodes ("Sampled Neighbors" in the figure).

age DINOv2 (x-axis, ↑) and FID score (y-axis, ↓) across the three datasets. INSTRUCTG2I outperforms most baselines on both metrics and achieves the best trade-off between them. InstructPix2Pix obtains a better FID score than INSTRUCTG2I because it takes an in-distribution image as input, constraining the output image to stay close to the original distribution.

**Qualitative Evaluation.** We conduct a qualitative evaluation by randomly selecting some generated cases. The results are shown in Figure 4, where we provide the sampled neighbor images from the graph, text prompts, and the ground truth images. From these results, we observe that INSTRUCTG2I generates images that best fit the semantics of the text prompt and context from the graph. For instance, when generating a picture for "the crater and the clouds", the baselines either capture only the content ("crater" and "clouds") without the style learned from the graph (Stable Diffusion and InstructPix2Pix) or adopt

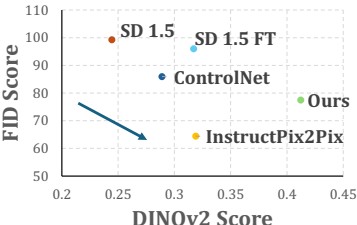

Figure 3: **INSTRUCTG2I achieves the best trade-off** between DINOv2 (↑) and FID (↓) scores.

a similar style but lose the desired content (ControlNet). In contrast, INSTRUCTG2I effectively learns from the neighbors on the graph and conveys the content accurately.

Table 2: Ablation study on graph condition variants and Graph-QFormer.

| Model | ART500K | | Amazon | | Goodreads | |
|---|---|---|---|---|---|---|
| | CLIP score | DINOv2 score | CLIP score | DINOv2 score | CLIP score | DINOv2 score |
| INSTRUCTG2I | **73.73** | **46.45** | **68.34** | **51.70** | **50.37** | **25.54** |
| - Graph-QFormer | 72.53 | 44.16 | 66.97 | 48.18 | 47.91 | 24.74 |
| + GraphSAGE | 72.26 | 43.06 | 66.07 | 43.40 | 46.68 | 21.91 |
| + GAT | 72.60 | 43.32 | 66.73 | 46.58 | 46.57 | 21.45 |
| IP2P w. neighbor images | 65.89 | 33.90 | 63.19 | 40.32 | 47.21 | 21.55 |
| SD FT w. neighbor texts | 69.72 | 38.64 | 65.55 | 43.51 | 47.47 | 22.68 |

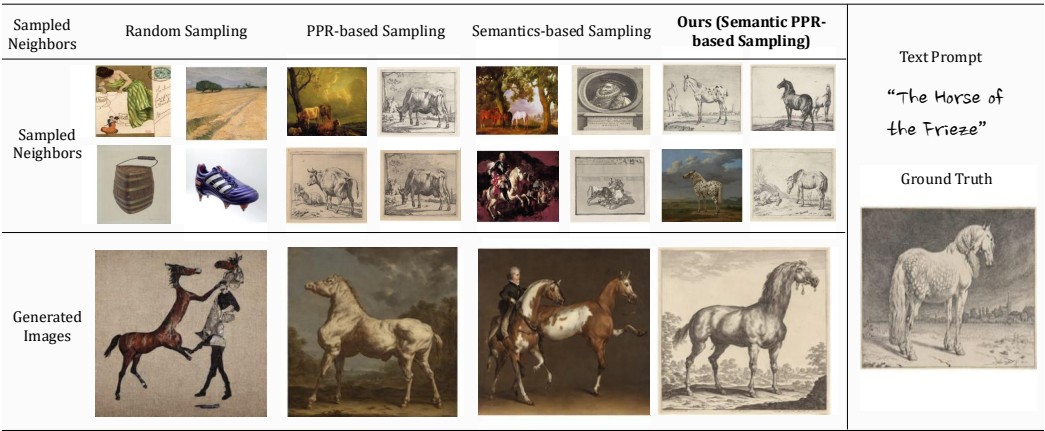

Figure 5: Ablation study on semantic PPR-based neighbor sampling. The results indicate that both structural and semantic relevance proposed by our method effectively improve the image generation quality and consistency with the graph context.

## 4.3 Ablation Study

**Study of Graph Condition for SD Variants.** In INSTRUCTG2I, we introduce graph conditions into SD by encoding the images from $c_G$ into graph prompts, which serve as conditions together with text prompts for SD's denoising step. In this section, we demonstrate the significance of this design by comparing it with other variants that utilize graph conditions in SD: InstructPix2Pix (IP2P) with neighbor images and SD finetuned with neighbor texts. For the first variant, we perform mean pooling on the latent representations of images in $c_G$, according to the IP2P's setting, and use this as the input image representation for IP2P. This variant has the same input information as INSTRUCTG2I. For the second variant, we utilize text information from neighbors instead of images, concatenate it with the text prompt, and fine-tune the SD. The results are shown in Table 2, where INSTRUCTG2I consistently outperforms both variants. This demonstrates the advantage of leveraging image features from $c_G$ and the effectiveness of our model design.

**Study of Graph-QFormer.** We first demonstrate the effectiveness of Graph-QFormer by replacing it with the simple baseline mentioned in Eq.(7), denoted as "- Graph-QFormer". We then compare it with graph neural network (GNN) baselines including GraphSAGE [13] and GAT [36], integrated into INSTRUCTG2I in the same manner. The results, presented in Table 2, show that INSTRUCTG2I with Graph-QFormer consistently outperforms both the ablated version and GNN baselines. This demonstrates the effectiveness of Graph-QFormer design.

**Study of the Semantic PPR-based Neighbor Sampling.** We propose a semantic PPR-based sampling method that combines structure and semantics for neighbor sampling on graphs, as detailed in Section 3.2. In this section, we demonstrate the effectiveness of this approach by conducting ablation studies that remove either or both components. The results, shown in Figure 5, indicate that our sampling methods effectively identify neighbor images that contribute most significantly to the ground truth in both semantics and style. This underscores the value of integrating both structural and semantic information in our sampling approach.

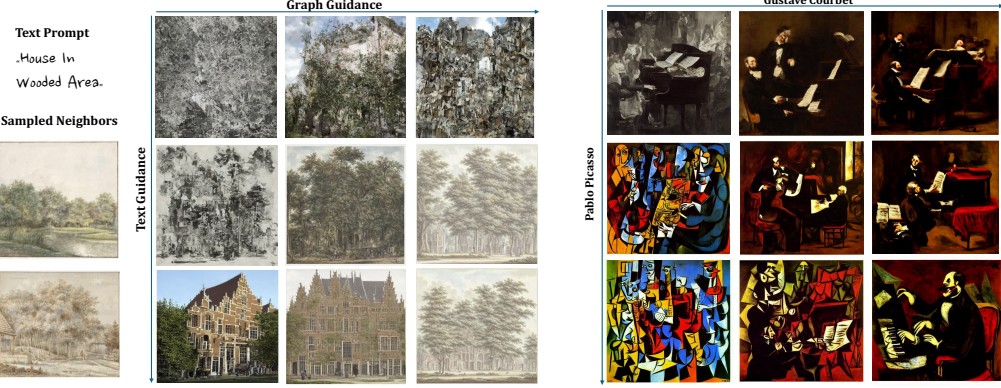

(a) Text and graph guidance study.          (b) Single or multiple graph guidance.

Figure 6: Controllable generation study. (a) The ability of INSTRUCTG2I to balance text guidance and graph guidance. (b) Study of multiple graph guidance. Generated artworks with the input text prompt "a man playing piano" conditioned on single or multiple graph guidance (styles of "Picasso" and "Courbet"). Please refer to Figure 1 for another example between Monet and Kandinsky.

## 4.4 Controllable Generation

**Text Guidance & Graph Guidance.**    In Eq.(11), we discuss the control of guidance from both text and graph conditions. To illustrate its effectiveness, we provide an example in Figure 6(a). The results show that as text guidance increases, the generated image incorporates more of the desired content. Conversely, as graph guidance increases, the generated image adopts a more desired style. This demonstrates the ability of our method to balance content and style through controlled guidance.

**Multiple Graph Guidance: Virtual Artist.**    In Eq.(12), we demonstrate how multiple graph guidance can be managed for controllable image generation. We present a use case, virtual artwork creation, to showcase its effectiveness (shown in Figure 6(b)). The goal of this task is to create an image that depicts specific content (*e.g.*, a man playing piano) in the style of one or more artists (*e.g.*, Picasso and Courbet). This is akin to adding a new node to the graph that links to the artwork nodes created by the specified artists and generating an image for this node. The results indicate that when single graph guidance is provided, the generated artwork aligns with that artist's style. As additional graph guidance is introduced, the styles of the two artists blend together. This demonstrates that our method offers the flexibility to meet various control requirements, effectively balancing different types of graph influences.

## 4.5 Model Behavior Analysis

**Cross-attention Weight Study in Graph-QFormer.** We conduct a cross-attention study for Graph-QFormer to understand how different sampled neighbors on the graph are selected based on the text prompt and contribute to the final image generation. We randomly select a case with the text prompt and neighbor images and plot the cross-attention weight map shown in Figure 7. From the weight map, we can find that Graph-QFormer learns to assign higher weight to pictures 1 and 4 which are related to "raising" and "Lazarus" in the text prompt respectively. The results indicate that Graph-QFormer effectively learns to select the images that are most relevant to the text prompt.

## 5 Related works

**Diffusion Models.** Recent advancements in diffusion models have demonstrated significant success in generative applications. Diffusion models [4, 7] generate compelling examples through a step-wise denoising process, which involves a forward process that introduces noise into data distributions and a reverse process that reconstructs the original data [19]. A notable example is the Latent Diffusion Model (LDM) [32], which reduces computational costs by applying the diffusion process

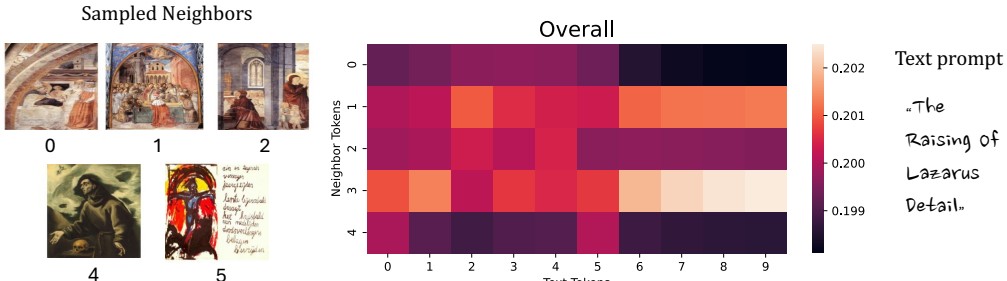

Figure 7: Study of Graph-QFormer's cross-attention map. Graph-QFormer effectively learns to select the images that are most relevant to the text prompt.

in a low-resolution latent space. In the domain of diffusion models, various forms of conditioning are employed to direct the generation process, including labels [6], classifiers [8], texts [28], images [2], and scene graphs [39]. These conditions can be incorporated into diffusion models through latent concatenation [33], cross-attention [1], and gradient control [12]. However, most existing works neglect the relational information between images and cannot be directly applied to image synthesis on MMAGs.

**Learning on Graphs.** Early studies on learning on graphs primarily focus on representation learning for nodes or edges based on graph structures [3, 14]. Methods such as Deepwalk [30] and Node2vec [11] perform random walks on graphs to derive vector representation for each node. Graph neural networks (GNNs) [38, 43] are later introduced as a learnable component that incorporates both initial node features and graph structure. GNNs have been applied to various tasks, including classification [25], link prediction [42], and recommendation [21]. For instance, GraphSAGE [13] employs a propagation and aggregation paradigm for node representation learning, while GAT [36] introduces an attention mechanism into the aggregation process. Recently, research has increasingly focused on integrating text or image features with graph structures [22, 44]. For example, Patton [23] proposes pretraining language models on text-attributed graphs. However, these existing works mainly target representation learning on single-modal graphs and are not directly applicable to the image synthesis from multimodal attributed graph (MMAG) task addressed in this paper.

# 6 Conclusions

In this paper, we identify the problem of image synthesis on multimodal attributed graphs (MMAGs). To address this challenge, we propose a graph context-conditioned diffusion model that: 1) Samples related neighbors on the graph using a semantic personalized PageRank-based method; 2) Effectively encodes graph information as graph prompts by considering their dependency with Graph-QFormer; 3) Generates images under control with graph classifier-free guidance. We conduct systematic experiments on MMAGs in the domains of art, e-commerce, and literature, demonstrating the effectiveness of our approach compared to competitive baseline methods. Extensive studies validate the design of each component in INSTRUCTG2I and highlight its controllability. Future directions include joint text and image generation on MMAGs and capturing the heterogeneous relations between image and text units on MMAGs.

## Acknowledgments and Disclosure of Funding

This work was supported by the Apple PhD Fellowship. The research also was supported in part by US DARPA INCAS Program No. HR0011-21-C0165 and BRIES Program No. HR0011-24-3-0325, National Science Foundation IIS-19-56151, the Molecule Maker Lab Institute: An AI Research Institutes program supported by NSF under Award No. 2019897, and the Institute for Geospatial Understanding through an Integrative Discovery Environment (I-GUIDE) by NSF under Award No. 2118329. Any opinions, findings, and conclusions or recommendations expressed herein are those of the authors and do not necessarily represent the views, either expressed or implied, of DARPA or the U.S. Government. The views and conclusions contained in this paper are those of the authors and should not be interpreted as representing any funding agencies.

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

# A Appendix

## A.1 Limitations

In this work, we focus on node image generation from multimodal attributed graphs, utilizing Stable Diffusion 1.5 as the base model for INSTRUCTG2I. Due to computational constraints, we leave the exploration of larger diffusion models, such as SDXL, for future work. Additionally, we model the graph as homogeneous, not accounting for heterogeneous node and edge types. Considering that different types of nodes and edges convey distinct semantics, future research could investigate how to perform *Graph2Image* on heterogeneous graphs.

## A.2 Ethical Considerations

While stable diffusion models [32] have demonstrated advanced image generation capabilities, studies highlight several drawbacks, such as the uncontrollable generation of NSFW content [9], vulnerability to adversarial attacks [45], and being computationally intensive and time-consuming [34]. In INSTRUCTG2I, we address these challenges by introducing graph conditions into the image generation process. However, since INSTRUCTG2I employs stable diffusion as the backbone model, it remains susceptible to these limitations.

## A.3 Classifier-free Guidance

In Section 3.4, we discuss controllable generation to balance text and graph guidances ($c_T$ and $c_G$) as well as managing multiple graph guidances ($c_G^{(k)}$). We introduce $s_T$ and $s_G$ to control the strength of text conditions and graph conditions and have the modified score estimation shown as follows (copied from Eq.(11)):

$$\hat{\epsilon}_\theta(\mathbf{z}_t, c_G, c_T) = \epsilon_\theta(\mathbf{z}_t, \varnothing, \varnothing) + s_T \cdot (\epsilon_\theta(\mathbf{z}_t, \varnothing, c_T) - \epsilon_\theta(\mathbf{z}_t, \varnothing, \varnothing))$$
$$+ s_G \cdot (\epsilon_\theta(\mathbf{z}_t, c_G, c_T) - \epsilon_\theta(\mathbf{z}_t, \varnothing, c_T)).$$

In this section, we will provide mathematical derivation on how these modified score estimations are developed. Noted that INSTRUCTG2I learns $P(\mathbf{z}|c_G, c_T)$, the distribution of image latents $\mathbf{z}$ conditioned on text information $c_T$ and graph information $c_G$, which can be expressed as:

$$P(\mathbf{z}|c_G, c_T) = \frac{P(\mathbf{z}, c_G, c_T)}{P(c_G, c_T)} = \frac{P(c_G|c_T, \mathbf{z})P(c_T|\mathbf{z})P(\mathbf{z})}{P(c_G, c_T)}. \tag{13}$$

INSTRUCTG2I learns and estimates the score [20] of the data distribution, which can also be interpreted as the gradient of the log distribution probability. By taking a log on both sides of Eq.(13), we can attain the following equation:

$$\log(P(\mathbf{z}|c_G, c_T)) = \log(P(c_G|c_T, \mathbf{z})) + \log(P(c_T|\mathbf{z})) + \log(P(\mathbf{z})) - \log(P(c_G, c_T)). \tag{14}$$

After calculating the derivation on both sides of Eq.(14), we can obtain:

$$\frac{\partial \log(P(\mathbf{z}|c_G, c_T))}{\partial \mathbf{z}} = \frac{\partial \log(P(c_G|c_T, \mathbf{z}))}{\partial \mathbf{z}} + \frac{\partial \log(P(c_T|\mathbf{z}))}{\partial \mathbf{z}} + \frac{\partial \log(P(\mathbf{z}))}{\partial \mathbf{z}}. \tag{15}$$

This corresponds to our classifier-free guidance equation shown in Eq.(11), where $s_T$ controls how the data distribution shifts toward the zone where $P(c_T|\mathbf{z})$ assigns a high likelihood to $c_T$ and $s_G$ determines how the data distribution leans toward the region where $P(c_G|c_T, \mathbf{z})$ assigns a high likelihood to $c_G$. Although there are other ways to derive the modified score estimation function (*e.g.*, switching $c_T$ and $s_G$ or making it symmetric), we empirically find that our derivation contributes to both advanced performance (since $P(c_T|\mathbf{z})$ is well learned in the base model) and high efficiency (since the denoising operation only needs to be conducted three times rather than four times compared with symmetric setting).

If given multiple graph conditions, we utilize $s_G^{(k)}$ to control the strength for each of them and have the derived score estimation function as follows (copied from Eq.(12)):

$$\hat{\epsilon}_\theta(\mathbf{z}_t, c_G, c_T) = \epsilon_\theta(\mathbf{z}_t, \varnothing, \varnothing) + s_T \cdot (\epsilon_\theta(\mathbf{z}_t, \varnothing, c_T) - \epsilon_\theta(\mathbf{z}_t, \varnothing, \varnothing))$$
$$+ \sum s_G^{(k)} \cdot (\epsilon_\theta(\mathbf{z}_t, c_G^{(k)}, c_T) - \epsilon_\theta(\mathbf{z}_t, \varnothing, c_T)).$$

Table 3: Dataset Statistics

| Dataset | # Node | # Edge |
|---------|--------|--------|
| ART500K | 311,288 | 643,008,344 |
| Amazon | 178,890 | 3,131,949 |
| Goodreads | 93,475 | 637,210 |

Table 4: Hyper-parameter configuration for model training.

| Parameter | ART500K | Amazon | Goodreads |
|-----------|---------|--------|-----------|
| Optimizer | AdamW | AdamW | AdamW |
| Adam $\epsilon$ | 1e-8 | 1e-8 | 1e-8 |
| Adam $(\beta_1, \beta_2)$ | (0.9, 0.999) | (0.9, 0.999) | (0.9, 0.999) |
| Weight decay | 1e-2 | 1e-2 | 1e-2 |
| Batch size per GPU | 16 | 16 | 16 |
| Gradient Accumulation | 4 | 4 | 4 |
| Epochs | 10 | 10 | 30 |
| Resolution | 256 | 256 | 256 |
| Learning rate | 1e-5 | 1e-5 | 1e-5 |
| Backbone SD | | Stable Diffusion 1.5 | |

If multiple graph conditions are given, Eq.(13) then becomes:

$$P(\mathbf{z}|c_G^{(1)}, ..., c_G^{(M)}, c_T) = \frac{P(\mathbf{z}, c_G^{(1)}, ..., c_G^{(M)}, c_T)}{P(c_G^{(1)}, ..., c_G^{(M)}, c_T)} = \frac{P(c_G^{(1)}, ..., c_G^{(M)}|c_T, \mathbf{z})P(c_T|\mathbf{z})P(\mathbf{z})}{P(c_G^{(1)}, ..., c_G^{(M)}, c_T)}, \quad (16)$$

where $M$ is the total number of graph conditions.

Assume $c_G^{(k)}$ are independent from each other, then we can attain:

$$P(\mathbf{z}|c_G^{(1)}, ..., c_G^{(M)}, c_T) = \frac{\prod_k P(c_G^{(k)}|c_T, \mathbf{z})P(c_T|\mathbf{z})P(\mathbf{z})}{P(c_G^{(1)}, ..., c_G^{(M)}, c_T)}. \quad (17)$$

Similar to Eq.(15), we can obtain:

$$\frac{\partial \log(P(\mathbf{z}|c_G^{(1)}, ..., c_G^{(M)}, c_T))}{\partial \mathbf{z}} = \sum_k \frac{\partial \log(P(c_G^{(k)}|c_T, \mathbf{z}))}{\partial \mathbf{z}} + \frac{\partial \log(P(c_T|\mathbf{z}))}{\partial \mathbf{z}} + \frac{\partial \log(P(\mathbf{z}))}{\partial \mathbf{z}}. \quad (18)$$

This corresponds to the classifier-free guidance equation shown in Eq.(12), where $s_G^{(k)}$ determines how the data distribution leans toward the region where $P(c_G^{(k)}|c_T, \mathbf{z})$ assigns a high likelihood to the graph condition $c_G^{(k)}$.

## A.4 Datasets

The statistics of the three datasets can be found in Table 3. Since Amazon and Goodreads both have multiple domains, we select one from each of them considering the graph size: Beauty domain from Amazon and Mystery domain from Goodreads.

## A.5 Experimental Settings

We randomly mask 1,000 nodes as testing nodes from the graph for all three datasets and serve the remaining nodes and edges as the training graph.

In implementing INSTRUCTG2I, we initialize the text encoder and U-Net with the pretrained parameters from Stable Diffusion 1.5 [1]. We use the pretrained CLIP image encoder as our fixed image encoder to extract features from raw images. For Graph-QFormer, we empirically find that initializing it with the CLIP text encoder parameters can improve performance compared with random initialization.

---

[1] https://huggingface.co/runwayml/stable-diffusion-v1-5

We use AdamW as the optimizer to train INSTRUCTG2I. The training of all methods including INSTRUCTG2I and baselines on ART500K and Amazon are conducted on two A6000 GPUs, while that on Goodreads is performed on four A40 GPUs. Each image is encoded as four feature vectors with the fixed image encoder following [40] and we insert one cross-encoder layer after every two self-attention layers in Graph-QFormer following [26]. The detailed hyperparameters are in Table 4.

