# OpenReview forum: "InstructG2I: Synthesizing Images from Multimodal Attributed Graphs"
_NeurIPS.cc/2024/Conference — NeurIPS 2024 poster_

### Official Review · Reviewer_7dPX · 2024-07-09

**Soundness:** 3
**Presentation:** 3
**Contribution:** 2
**Rating:** 5
**Confidence:** 3

**Summary:**

The authors propose an approach to enhance image synthesis using multimodal attributed graphs, adopting a strategy to condition image generation via a tokenization scheme on graph structure.

**Strengths:**

- The paper studies an intersectional topic: leveraging graph learning techniques for image generation, which is a creative application and an area which deserves more focus.
- The authors' use of qualitative examples (e.g. Figure 5 and 6) is commendable and helps articulate visual improvements.

**Weaknesses:**

Please see questions and concerns below.  My general feeling is the paper is fairly incremental in its introduction of a mechanism to encode graph condition into the conditioning for generation.  Many design choices for graph conditioning are not discussed well and the quantitative results for some of these choices are missing which hurts the overall impact of the work.

**Questions:**

- Typos:
  - Line 18: "graph-structued"

- The motivation proposed in lines 28-30 is a little bit confusing, since the scenario the authors discuss here (e.g. virtual artwork creation based on nuanced styles of artists and genres) seems like it could be well-handled by text rather than explicitly using graph structure.

- There is limited prior work in multimodal graph learning, as the authors mention.  The authors may want to reference and position their work with respect to the recent [1] which offers multiple datasets and focuses on utility of GNN methods for node/link/graph-level tasks rather than generative tasks.

- Nit: the notation is a bit awkward compared to conventional graph literature which typically uses $\mathcal{V}, \mathcal{E}, \mathcal{X}, \mathcal{F}$ or something similar to indicate node-set, edge-set, node-features, and edge-features.  The authors proposed notation in line 72 seems to define P and D as different sets of images / documents compared to the nodes V, but then mentions that each node has some textual information and image information corresponding to P and D (it should be made clear whether this information is just features, or actual node relationships -- if the latter, it seems that P and D should be contained within the nodeset V).

- The process described in line 115 around introducing tokens to help condition the representation using graph structure is also explored in some related works, e.g. [2].  Perhaps the authors could consider adopting a similar approach if it makes sense in this task, since the tokenization scheme as the authors of [2] point out is key in injecting the right level of information to the model.

- Comment: the notation in pages 3-5 is quite heavy and would benefit from a symbol table.

- Section 3.2 proposes a heuristic solution for neighbor selection.   I'd encourage exploring solutions designed for learnable importance of multiple edge types similar to [3].

- Can the authors discuss what sort of techniques were used to incorporate graph conditions for the baseline models like InstructPix2Pix and SD?

- Is there a quantitative understanding or experiment for the PPR neighbor based sampling approach?  It seems this is one of the more heuristic parts of the paper where the design of the sampling procedure (two phase PPR + semantic re-ranking) is less conventional and deviates from other aggregation mechanisms explored in previous literature like attention-based selection, learnable importance of multiple edge types, etc.  The qualitative experiment is helpful but not terribly convincing in terms of the actual performance impact in aggregate.

[1] Multimodal Graph Benchmark (Zhu et al, 2024)

[2] LLaGA: Large Language and Graph Assistant (Chen et al, ICML 2024)

[3] Pathfinder Discovery Networks for Neural Message Passing (Rozemberczki et al, WWW 2021)

**Limitations:**

Yes, Appendix A.1

---

> ### Author Rebuttal · Authors · 2024-08-06
>
> Thank you so much for your thoughtful review!
>
> Regarding your questions:
>
> 1. **The scenario the authors discuss in lines 28-30 seems like it could be well-handled by only text.**
> We would like to answer this question from three aspects. 1) On one hand, the problem introduced in this paper can be grounded not only in the virtual artwork scenario but also in others such as the e-commerce scenario, where generating an image for a product node connected to other products equates to recommending future products. Such scenarios are hard to handle by text only (e.g., user interests are implicitly expressed in their purchase history which is hidden in the graph structure). 2) On the other hand, even in the artwork scenario, for the artists who are not that famous (e.g., my little brother), it is hard to use text to represent them; and for the artists who are new and not seen during model training, it is hard to expect the model can be generalized through text (artist names). However, using graph structure as the condition, we can well solve the not famous artist and new artist problem by discovering neighbors in the graph. We add an example of controllable generation for a node connecting to “Pablo Picasso” and my little brother on the artwork graph to illustrate this in the rebuttal PDF (in general response). 3) Another advantage of adopting graph conditions is that we can combine multiple graph conditions (e.g., different art styles) with controllable generation (with mixed ratio according to the user’s interest), which is extremely hard to express by text only.
>
> 2. **Add a reference to the recent [1].**
> Thank you for bringing this related work. We would like to mention that the reason why we do not reference this paper [1] in our submission is that *it is put on arxiv later than the neurips submission deadline*. However, as the reviewer mentioned and we agreed, this is a related paper and we would like to reference it in our revision.
>
> 3. **Question about the notations, especially P, D, and V.**
> We agree with the reviewer that since this paper tackles multimodal learning on graphs, the notations will be a little different and complex compared with conventional graph literature. As mentioned in the paper, each node $v_i\in V$ is associated with a text $d_{v_i}\in D$ and an image $p_{v_i}\in D$. In other words, $d_{v_i}$ and $p_{v_i}$ can be understood as features for $v_i$ from the conventional graph learning perspective. We will make it clear in the revision.
>
> 4. **Perhaps the authors could consider adopting a similar approach [2] if it makes sense in this task.**
> Thank you for bringing up this great paper and we are glad to reference it in the revision. At the same time, we would like to answer this question from three aspects: 1) *The design of Graph-QFormer is non-trivial*: as we mentioned in the paper, in MMAGs, we need to tackle the graph entity dependency (including image-image dependency and text-image dependency) to conduct image generation. To extract such dependency, we design Graph-QFormer and show consistently better performance than other designs in Table 2; 2) *Different task with different backbone models*: we agree with the reviewer that LLaGA is a great work, but we would like to emphasize that the problem tackled in LLaGA (text generation on TAGs with LLM) is different from that in this paper (image generation on MMAGs with stable diffusion). Since the problem and the backbones are different, it is non-trivial to directly compare these two methods; 3) We have compared with simple GNN tokenization methods in the ablation study (Table 2) and demonstrate the effectiveness of Graph-QFormer compared with them.
>
> 5. **Comment: the notation on pages 3-5 is quite heavy and would benefit from a symbol table.**
> We will add a symbol table in either the main content or the appendix in the revision according to your suggestion.
>
> 6. **Learnable importance of multiple edge types similar to [3].**
> We agree with the reviewer that learnable sampling is a promising future direction. As we are the first paper to introduce this problem, we would like to tackle this problem in a simple and effective way. As a result, we would like to leave this more complex learnable sampling for future studies.
>
> 7. **How graph conditions are introduced to InstructPix2Pix and SD?**
> We compare our method with InstructPix2Pix and SD with an advanced design for graph information in Table 2. For InstructPix2Pix, we aggregate the neighboring images on the graph with mean pooling and serve the resulting image as the input image condition. For SD, we concatenate the text information from the center nodes’ neighbors on the graph to its original text as text condition. From Table 2, our method outperforms both baselines significantly which demonstrates the effectiveness of the InstructG2I design.
>
> 8. **A quantitative understanding of semantic PPR-based neighbor sampling approach.**
> We are glad to add some quantitative results (DINOv2 score) on ART500K and Amazon datasets as below:
>
> | Dataset             | ART500K | Amazon |
> |---------------------|--------|--------|
> | Ours  | 46.45 | 51.70 |
> | - PPR | 45.06 | 48.40 |
> | - semantic ranking | 46.19 | 51.49 |
>
> From the result, we can see that both PPR-based sampling and semantic-based reranking can benefit the InstructG2I. This is demonstrated in both quantitative and qualitative results (Figure 5). The reason we adopt semantic PPR-based sampling rather than attention-based selection and others is that semantic PPR-based sampling can be performed offline only once and it is computationally efficient compared with SD training and inference. This enables us to scale to large-scale graphs in the real world. However, we agree with the reviewer that attention-based selection and others are interesting directions and we would like to leave them for future research.

---

> ### Comment · Reviewer_7dPX · 2024-08-12
> **Thank you**
>
> Dear authors -- thank you for your response to my concerns.  I think adding this discussion to the work will help strengthen the positioning (especially the parts around necessity/utility of graph conditioning vs. text).  Overall, I stand by my initial review that parts of the work feel a bit heuristic and incremental, but the work is generally an interesting proposal which leans towards a different application of graph structure than most are used to seeing. I will retain my score.

---

> > ### Author Response · Authors · 2024-08-13
> >
> > Dear Reviewer 7dPX,
> >
> > Thank you so much for your reply! We will add those discussions to the revision according to your suggestions.
> >
> > We would like to emphasize the contributions of the work again here:
> > - (**Problem**). We are pioneers in recognizing the potential of multimodal attributed graphs (MMAGs) for image synthesis and introducing the Graph2Image problem.
> > - (**Algorithm**). We introduce InstructG2I, a context-aware diffusion model that adeptly encodes graph conditional information as graph prompts for controllable image generation.
> > - (**Benchmark**). We construct a benchmark with graphs from three domains (art, e-commerce, literature) to evaluate the models on the Graph2Image problem. The benchmark can be used for future exploration of this problem.
> > - (**Experiments**). We perform experiments on the benchmark, showing that InstructG2I consistently surpasses competitive baselines.
> >
> > For the (**Algorithm**) part, we propose four main novel components:
> > - *Semantic PPR-based neighbor sampling*: It helps discover the informative neighbors based on semantic and structure information for target node image generation.
> > - *Graph-QFormer*: It extracts the common features of the selected neighbors and aggregates the neighbor information as graph conditional tokens for the stable diffusion model, considering the image-image dependency and text-image dependency.
> > - *Graph conditions as tokens*: It enables stable diffusion to leverage the graph condition information for target image generation.
> > - *Graph-based controllable generation (Graph CFG)*: It enables controllable, tunable generation with multiple graph conditions.
> >
> > Although the philosophy of “graph tokens” is used in graph-enhanced text generation with LLMs, it is not explored in graph-enhanced image generation with diffusion models. Besides, it is nontrivial to get those graph tokens (semantic PPR-based neighbor sampling and Graph-QFormer) and use them for controllable generation (graph CFG).
> >
> > We would like to argue that even though the reviewer insists that “graph tokens” are incremental, it is only a small part of our **Algorithm** designs. We have other novel design components in **Algorithm** and other contributions in addition to **Algorithm** (**Problem**, **Benchmark**, **Experiments**).
> >
> > We would like to thank the reviewer again for the reply. If you have any more thoughts, we are happy to continue our discussion until the deadline.

---

### Official Review · Reviewer_nWct · 2024-07-12

**Soundness:** 3
**Presentation:** 2
**Contribution:** 3
**Rating:** 6
**Confidence:** 3

**Summary:**

This paper focuses on the problem of image synthesis on multimodal attributed graphs (MMAGs) and proposes a graph context-conditioned diffusion model, INSTRUCTG2I, to address the challenge in this setting. In particular, it proposes a semantic personalized PageRank-based method to sample related neighbors in the graph.  Then, the INSTRUCTG2I can effectively encode graph conditional information as graph prompts with Graph-QFormer. Systematic experiments on MMAGs demonstrate the effectiveness of the methods proposed in this paper compared to competitive baseline methods.

**Strengths:**

1.  This paper studies an interesting and meaningful question. It investigates the graph-structured relationships of real-world entities for image generation on MMAGs, a task well-grounded in practical applications.
2. This paper is well-structured and easy to understand.
3. The graph context-conditioned diffusion model proposed in this paper is reasonable in solving image generation problems on MMAGs.

**Weaknesses:**

1. The description in eq.10 may be incorrect. Please check more carefully.
2. Subsection 3.4 is more challenging to understand when reading. The authors' descriptions of some symbols in Eq. 10 and Eq. 11 are not exhaustive.
3. The results of the ablation experiments in Table 2 indicate that using a GNN such as GAT or GraphSAGE to aggregate graph information seems to be worse than the straightforward approach in Eq.7. Authors are requested to give a more detailed discussion with a reasonable explanation.
4. The images sampled by the semantic PPR-based sampling shown in Figure 5 appear to have the same image as the ground truth. Does this indicate that the proposed method suffers from label leakage?

**Questions:**

1. Please see the weaknesses.
2. I wonder if the authors will compare it to other state-of-the-art image generation models, such as some Multimodal LLMs that are so prevalent nowadays.

**Limitations:**

This paper has reasonably discussed the limitations.

---

> ### Author Rebuttal · Authors · 2024-08-06
>
> Thank you so much for your thoughtful review!
>
> Regarding your questions:
>
> 1. **The description in Eq.10 may be incorrect.**
> Thank you so much for your comment. We have found the typos and will correct them in the revision.
>
> 2. **Descriptions of symbols in subsection 3.4.**
> This section mainly discusses controllable generation with classifier-free guidance (CFG) on graphs. The main philosophy of CFG is that $\epsilon_\theta$ learns the gradient of the log image distribution, and increasing the contribution of $\epsilon_\theta(c,z_t)-\epsilon_\theta(z_t)$ will enhance convergence towards the distribution conditioned on $c$. We agree with the review that the description of the symbols can be improved to make this section easier to understand. We plan to add detailed illustrations to symbols including $\varnothing$, $z_t$, $s^{(k)}_G$, and others. We would be grateful if the reviewer could provide us with inputs on what other symbols need further explanation and we are glad to make this section clearer in the revision.
>
> 3. **Why do GNNs underperform the straightforward approach in Eq.7?**
> Thank you for the comments. The philosophy of vanilla GNN is to propagate and aggregate information from neighbors and compress the neighboring information into *one* embedding for the center node, while the straightforward approach in Eq.7 will directly provide neighboring information separately without aggregation or compression. It is worth noting that the aggregation/compression step could lead to neighbor information loss, while the straightforward approach in Eq.7 does not. Since more information is provided to the stable diffusion model, the latter can perform better for image generation. However, we believe that future work can consider a more advanced design of GNN methods that could keep all the neighboring information and extract the common knowledge for more accurate image generation on graphs.
>
> 4. **The images sampled by the semantic PPR-based method appear to have the same image as the ground truth. Does this indicate any label leakage?**
> We are sorry for the confusion. This is a typo and we will correct it in our revision. We have written data processing codes to ensure that there is no label or data leakage in both training and testing. To demonstrate this, we have uploaded the data processing code here: https://anonymous.4open.science/r/Graph2Image-submit-607E/art500k_dp.ipynb
>
> 5. **Comparison with other SOTA multimodal LLMs.**
> Thank you for your comments. We would like to argue that the philosophy of introducing graph information into image generation can be not only adopted to the diffusion model (in our paper) but also to any multimodal LLM backbones as mentioned by the reviewer. However, we agree with the reviewer that adding the comparison can make the experiment more comprehensive. As a result, we compare with a recent advanced multimodal LLM called DreamLLM [1] and show the results below:
>
> The image-image CLIP score:
>
> | Dataset             | ART500K | Amazon | Goodreads |
> |---------------------|--------|--------|------|
> | DreamLLM  | 57.16 | 60.72 | 40.72 |
> | Ours | 73.73 | 68.34 | 50.37 |
>
> The image-image DINOv2 score:
>
> | Dataset             | ART500K | Amazon | Goodreads |
> |---------------------|--------|--------|------|
> | DreamLLM  | 27.06 | 34.77 | 16.13 |
> | Ours | 46.45 | 51.70 | 25.54 |
>
> From the results, we can find that our method can outperform the advanced DreamLLM consistently, which demonstrates the effectiveness of our model design.
>
> [1] DreamLLM: Synergistic Multimodal Comprehension and Creation. ICLR 2024.

---

> > ### Comment · Reviewer_nWct · 2024-08-11
> >
> > Thank you for your detailed response, the author effectively addressed my issue, I will increase my score.

---

> ### Author Response · Authors · 2024-08-11
>
> Dear Reviewer nWct,
>
> We are glad that we address your raised issues and we are grateful that you increase the score!
>
> We will further enhance the submission according to your suggestions.
>
> Best,
>
> Authors

---

### Official Review · Reviewer_5RsF · 2024-07-13

**Soundness:** 3
**Presentation:** 3
**Contribution:** 4
**Rating:** 7
**Confidence:** 3

**Summary:**

The paper introduces a new task graph2image which is to generate images conditioned on both text descriptions and graph information, which improves consistency of generated images compared to conditioned only on texts or images. To address combinatorial complexity of graphs and dependencies among graph entities, the paper proposes a graph context-conditioned diffusion model InstructG2I for generating images from multimodal attributed graph.

**Strengths:**

- To the best of my knowledge, graph2image is a novel task, and the motivation to use the rich and high-dimensional information of graphs for image generation seems reasonable and interesting.
- The proposed approach to incorporate graph information into pre-trained text-to-image is new, in particular introducing graph conditioning token and considering scalability of graph size.
- The generated samples show that using graph information results in better consistency with the ground truth compared to methods that use only text prompts or images.
- Examples of controllable generation with both text and graph show the ability to balance content and style in a simple manner.

**Weaknesses:**

While I do not have a major concern, an ablation study on scalability to graph size seems to be missing. How large graphs is the method able to be applied?

**Questions:**

- Why is the DINOv2 score on Goodreads dataset significantly low compared to that of ART500K or Amazon datsets?

**Limitations:**

Yes the paper addresses the limitation in Appendix A.1.

---

> ### Author Rebuttal · Authors · 2024-08-06
>
> Thank you so much for your thoughtful review and support of our work!
>
> Regarding your questions:
>
> 1. **How large graphs can the method be applied?**
>
> Thank you so much for your question. Our method can be adapted to large-scale graphs with millions or even trillions of nodes. In InstructG2I, we only need to conduct offline sampling with semantic PPR-based neighbor sampling **once** to extract useful structure information from the large-scale graphs. Since this step is performed offline, the size of the graph will not introduce sampling bottlenecks to stable diffusion model training and inference. Empirically, the semantic PPR-based neighbor sampling only takes about 10 minutes on a graph with millions of nodes, which is quite short compared to stable diffusion training and inference time cost.
>
> 2. **Why is the DINOv2 score on the Goodreads dataset lower than the others?**
>
> Thank you for your great observation. 1) *Training data size*: As shown in Table 3, the graph size of Goodreads is the smallest compared with ART500K and Amazon. This means that the training data on Goodreads is smaller than the other two datasets. Since a larger training data size can contribute to more sufficient training, the results on ART500K and Amazon are better than the results on Goodreads. 2) *Data distribution*: In ART500K, Amazon, and Goodreads datasets, the images are art pictures, product pictures, and book cover pictures respectively (some samples can be found in the rebuttal PDF in general response). We believe that compared with book cover pictures, product and art pictures are closer to the distribution of images used in stable diffusion (which is our base model) pretraining. In conclusion, training InstructG2I on ART500K and Amazon will provide better performance than training InstructG2I on the Goodreads dataset.

---

> > ### Comment · Reviewer_5RsF · 2024-08-11
> >
> > Thank you for the responses. I believe this paper tackles an interesting task with novel approach, and do not find any major concerns. Thus I will keep my score.

---

> > > ### Author Response · Authors · 2024-08-11
> > >
> > > Dear Reviewer 5RsF,
> > >
> > > Thank you for your continuous support of our work! We will further enhance the submission according to your suggestions.
> > >
> > > Best,
> > >
> > > Authors

---

### Official Review · Reviewer_TQRx · 2024-08-05

**Soundness:** 2
**Presentation:** 3
**Contribution:** 3
**Rating:** 5
**Confidence:** 3

**Summary:**

This paper introduces a novel approach for controllable image generation using both graph and text conditions. The authors propose that additional context information from multimodal attributed graphs (MMAGs) can enhance the performance of diffusion models. Specifically, they formulate the Graph2Image problem and develop the INSTRUCTG2I model to incorporate contextual information during the generation process. Empirical evaluations demonstrate the strong performance of the model.

**Strengths:**

1. The paper is easy to follow.
2. The intuition behind the approach is clear.

**Weaknesses:**

1. The overall setting is questionable. The authors integrate graph information using a Graph-QFormer and context information such as artists and genres, stored in graph prompt tokens. Given the large graph size, they only use subgraph structures. Consequently, the Stable Diffusion (SD) model absorbs additional information from similar artworks, which could be derived from image or text prompts alone. This raises the question of whether an additional condition structure is necessary. I suggest the authors demonstrate a unique application where standard models with text and image prompting capabilities are insufficient.

**Questions:**

1. Are there any unique scenarios where only graph input can significantly improve SD performance?

As my review is overdue, I welcome concise feedback and am open to clarifying any potential misunderstandings.

---

> ### Author Rebuttal · Authors · 2024-08-06
>
> Thank you so much for your thoughtful review!
>
> Regarding your questions:
>
> 1. **Question about the problem setting.**
>
> Thank you for your comment. We would like to answer this from two aspects.
>
> **Why graph is important?**
>
> 1) *Graph structure helps discover multiple informative neighbors*: We agree with the reviewer that Graph-QFormer is to transfer “neighbor images” into graph prompt tokens for center node generation. However, how to select such “neighbor images” is important and non-trivial, which needs help from graph structure. As shown in Figure 5, the “neighbor images” condition affects the generation results a lot, and our proposed semantic PPR-based sampling can discover informative neighbors based on *graph information* for high-quality image generation.
>
> 2) *We utilize global graph structure and semantics*: We would like to clarify that in our method, we do not “only use subgraph structures” but utilize global graph structure as well as text semantics for neighbor sampling. In semantic PPR-based sampling, we first adopt PPR on the *global graph* to discover informative neighbors based on *graph structure* and then adopt semantic search to find more fine-grained informative neighbors (e.g., if we would like to generate a picture of a “horse”, the neighbors related to “horse” will be more useful, as shown in Figure 5). In addition to the qualitative result, we also show the quantitative analysis of semantic PPR-based sampling, where we can find that utilizing global graph structure (Ours in the table below) outperforms utilizing only subgraph structures (- PPR in the table below), which also demonstrates the importance of graph structure information.
>
> | Dataset             | ART500K | Amazon |
> |---------------------|--------|--------|
> | Ours  | 46.45 | 51.70 |
> | - PPR | 45.06 | 48.40 |
> | - semantic ranking | 46.19 | 51.49 |
>
> 3) *Not just one, but many and extract their similarity from graph*: Graph enables discovering *multiple* related neighbor images and extracting their similarity rather than solely based on one image to conduct image generation (which is widely adopted in the literature). This is important for image generation in many scenarios. For example, we would like to generate a “bird” picture drawn by “Monet”. If we only conditioned on his picture of a “scenery”, we may overfit the corresponding content and fail to draw a “bird”. However, if we conditioned on a multiple of his images covering diverse content including animals, the model would better extract his style and successfully conduct image generation.
>
> **Why are image or text prompts alone not enough?**
>
> 1) *Text is not enough to describe everything*: For example, in the artwork scenario, for the artists who are not that famous (e.g., my little brother), it is hard to use text to represent them; and for the artists who are new and not seen during model training, it is hard to expect the model can be generalized through text (artist names). However, using graph structure as the condition, we can well solve the not famous artist and new artist problem by discovering informative neighbors in the graph. We add an example of image generation for my little brother on the artwork graph to illustrate this in the rebuttal PDF (in general response).
>
> 2) *Image conditions need to be discovered from the graph*: As discussed above, neighboring image conditions can affect the generation result a lot, and graph structure can benefit the informative neighbor image sampling. In addition, solely based on image condition will result in unclear content (e.g., if we want to generate a “bird” image connected to the Picasso node, solely based on the sampled neighboring images does not provide the content indicator of a “bird”).
>
> 3) *Graph enables flexible and diverse controllable generation*: Another advantage of adopting graph as the condition is that we can combine multiple graph conditions (e.g., different art styles) with the controllable generation (with any mixed ratio according to the user’s interest), which is extremely hard to express by text only or image only. We add an example of controllable generation for a node connecting to “Pablo Picasso” and my little brother on the artwork graph to illustrate this in the rebuttal PDF (in general response).

---

### Author Rebuttal · Authors · 2024-08-06

Dear Reviewers,

We sincerely appreciate your valuable feedback and suggestions. We will revise our work based on your reviews.

We also want to thank the Reviewers for noting the strengths of our paper, namely:

- The problem addressed in our paper is important and well-motivated. (5RsF, nWct, 7dPX)
- Our proposed method is substantial and modern. (TQRx, 5RsF, nWct)
- The paper is clearly written. (TQRx, nWct)
- The empirical results are consistent, solid, and convincing. (TQRx, 5RsF, 7dPX)
- The method offers balanced controllable generation. (5RsF)

We have addressed the individual questions of reviewers in separate responses. In the revised version, we will incorporate all reviewers' suggestions by making the motivation more solid, adding more experimental results/discussion, adding references, and making the symbols more clear. **We have also attached a rebuttal PDF below.**

Here we would like to briefly outline the contribution of this work for the reference of reviewers to start the discussion.

- (Formulation and Benchmark). We are pioneers in recognizing the potential of multimodal attributed graphs (MMAGs) for image synthesis, and we have introduced the Graph2Image problem. Our formulation is validated by three benchmarks based on practical applications in art and e-commerce. Those benchmarks will be valuable for future research.
- (Algorithm). Methodologically, we introduce InstructG2I, a context-aware diffusion model that adeptly encodes graph conditional information as graph prompts for controllable image generation (as illustrated in Figure 1(b,c) and the experiment section).
- (Experiments and Evaluation). Empirically, we perform experiments using graphs from three distinct domains, showing that InstructG2I consistently surpasses competitive baselines (as illustrated in Figure 1(b) and the experiment section).

In closing, we thank the Reviewers again for their time and valuable feedback. If there are further concerns, please let us know, and we will be happy to address them.

---

### Comment · Area_Chair_C5pZ · 2024-08-11
**reply to author's response**

Dear reviewers,

Could you take time to read the author's rebuttal and submit your feedback?
Thank you.

AC

---

### Decision · Program_Chairs · 2024-09-25

**Decision:**

Accept (poster)

**Comment:**

The paper introduces the Graph2Image problem and develops the InstructG2I model to incorporate graph structure and multimodal information for better image generation. Empirical evaluations demonstrate the strong performance of the model. The paper is well-written and easy to follow. Reviewers are generally positive about this work. They agree that the paper studies a novel task and proposes a reasonable approach. The authors are suggested to address reviewers' comments, e.g., clearer motivation discussion and notations, for paper revision.

Based on the above summary, I recommend acceptance.